# Glass-Forming Ability and Thermal Properties of Al_70_Fe_12.5_V_12.5_X_5_(X = Zr, Nb, Ni) Amorphous Alloys via Minor Alloying Additions

**DOI:** 10.3390/nano11020488

**Published:** 2021-02-15

**Authors:** Xuan Liu, Xingfu Wang, Yongli Si, Fusheng Han

**Affiliations:** 1Institute of Solid State Physics, Hefei Institutes of Physical Science, Chinese Academy of Sciences, Hefei 230031, China; liuxuan@ahjzu.edu.cn (X.L.); wangxingfu@issp.ac.cn (X.W.); siyongli@mail.ustc.edu.cn (Y.S.); 2Scinece Island Branch, Graduate School of USTC, Hefei 230026, China; 3Prefabricated Building Research Institute of Anhui Province, Anhui Jianzhu University, Hefei 230061, China

**Keywords:** amorphous alloys, glass-forming ability, thermal analysis, Al, mechanical alloying

## Abstract

The Al_70_Fe_12.5_V_12.5_Ni_5_, Al_70_Fe_12.5_V_12.5_Zr_5_ and Al_70_Fe_12.5_V_12.5_Nb_5_ alloys were prepared via mechanical alloying. The influence of Zr, Nb or Ni addition on the glass-forming ability of Al-Fe-V amorphous alloys have been investigated. The structure of Al_70_Fe_12.5_V_12.5_Ni_5_ was amorphous and Al_70_Fe_12.5_V_12.5_Zr_5_ was not completely amorphous by transmission electron microscopy, selected area electron diffraction and differential scanning calorimetry. Different criteria were used to evaluate the influence of the addition of alloy elements on the Glass-forming ability. The Al_70_Fe_12.5_V_12.5_Ni_5_ amorphous alloys exhibits higher glass-forming ability and activation energies of crystallization. Comparison of the effective atomic size ratio and mixture enthalpy on the glass-forming ability of these amorphous alloys demonstrates that the effective atomic size ratio value becomes more significant than the values of mixture enthalpy.

## 1. Introduction

Al-based amorphous alloy are of great interest due to their high specific strength, low density, low elastic modulus, good corrosion resistance and practically used metals [1,2,3,4]. The pressing and sintering of Al-based amorphous alloy powders are important methods to prepare bulk amorphous and nanomaterials, which is conducive to expand its application prospect in aerospace and other fields. The hot working of amorphous powder needs better thermal stability and higher glass-forming ability (GFA), which needs to produce an excellent amorphous powder. Researchers have been a focused effort that aims to understand the effect of minor alloying additions on GFA and thermal stability and have proposed many theories on structure, thermodynamic and kinetic factors, so as to find out the rule of amorphous formation, optimize the composition design of amorphous alloy and find the amorphous alloy system which are beneficial to later application [5,6,7,8,9,10,11,12,13,14]. It is generally believed that the amorphous alloy system should contain three or more elements, the atomic size difference of the main components should generally be greater than 12% and with a large negative mixing enthalpy. The structure of amorphous alloys is topological short-range order and the icosahedral fivefold packing is a more realistic ordering pattern of cluster–cluster connection in amorphous alloys. The type of Kasper polyhedra is controlled by the effective atomic size ratio between the solute and solvent atoms [7]. Therefore, the addition of minor alloy elements can change the effective atomic radius ratio, effectively increase the main underlying topological short-range order and then improve the glass forming ability of the system.

Historically, the glass-forming ability of Al-base system is low and it is difficult to form amorphous phase even by the mechanical alloying. With the development of amorphous theory, the researchers have found the addition of minor elements in the Al-base system can improve the amorphous glass forming ability (GFA) and thermal stability.

The GFA of Al-base base alloys can be improved by addition of Zr, Nb, Ni and other elements [15,16,17,18,19,20,21]. Bo Zhu et al. analyzed the addition of V, Fe and Cu to amorphous Al-base alloy and found that the GFA of Al_75_V_12.5_Fe_12.5-X_Cu_X_ increased with the decrease of Cu content due to the positive mixing enthalpies of V-Cu (+5 kJ/mol) and Fe-Cu (+13 kJ/mol) [21]. Although further addition of Cu could reduce the GFA of Al_75_V_12.5_Fe_12.5-X_Cu_X_ alloys, the beneficial effect of the addition of V, Fe is also recognized.

In the previous research work, we found that the addition of Nb in the Al-V-Fe amorphous alloy effectively enhances its glass forming ability and thermal stability [22]. The atomic size of the constituents varied in the order of Zr: 0.162 nm > Nb: 0.143 nm = Al: 0.143 nm > V: 0.132 nm > Ni: 0.125 nm > Fe: 0.124 nm [5]. Clearly, the types of coordination polyhedra in these various systems are difference owing to the different atomic size ratios. Hence, we consider adding Ni and Zr to Al-Fe-V alloy to form Al_70_Fe_12.5_V_12.5_Ni_5_ (Al-Ni), Al_70_Fe_12.5_V_12.5_Zr_5_ (Al-Zr) amorphous alloys, which are compared with Al_70_Fe_12.5_V_12.5_Nb_5_ (Al-Nb) amorphous alloys. We set out to understand the effect of minor alloying additions on GFA and thermal stability and to provide deeper insights into the development of Al-base amorphous alloys.

## 2. Experimental Details

Nominal composition of Al-Ni, Al-Zr and Al-Nb alloys were prepared via mechanical alloying. The purity of the powders were Al (99.9%), Fe (99.8%), V (99.9%), Zr(99.5%), Nb (99.5%), Ni (99.5%). According to the nominal composition of the amorphous alloys, the mixture of elemental powders (20 g) were weighed in respective proportions and blended with stearic acid (2 wt%). Stearic acid as the process control agent was used to restrict agglomeration and welding of the powders during mechanical alloying. The mixtures were put into a stainless steel cans (500 mL) with steel ball bearing (GGr15 with diameter of 20 mm, 10 mm and 5 mm are mixed) and ball-to-powder weight ratio of 20:1. In order to prevent these powder mixtures be oxidized the steel via be filled with argon in the vacuum glove box and then sealed and removed to high-energy planetary ball mill (QM-3SP4). The milling process was paused every 10 h to take a small amount of powder out of the cans for analysis to determine the milling time. The mechanical alloying (MA) was perform with the rotation speed of 400 rpm. X-ray diffraction (XRD, X’ Pert Pro MPD, PANalytical, Holland) was performed using Cu Kα (λ = 0.154 nm) radiation and transmission electron microscopy (TEM, JEM- 2010, JEOL, Japan) and selected area electron diffraction (SAED) to investigate the structure of powder mixtures samples. Highscore Plus (Philips, Holland) software program was used for processing of the XRD spectra. The thermal parameters were examined via the differential scanning calorimetry (DSC, NETZSCH DSC 404F3, Germany) under a continuous flow of Ar atmosphere.

## 3. Results and Discussion

Figure 1 shows the XRD patterns of Al-Zr, Al-Nb and Al-Ni alloys. Among the three alloying systems, all the typical diffraction peaks for crystalline phases disappeared and broad diffuse diffraction peaks of amorphous appeared. With the decrease of atomic size of additive alloy elements, the angular position of the diffuse maxima in the XRD pattern towards larger angles and the ball milling time required for amorphous decreases. The 2θ position of main diffuse maxima of Al-Zr, Al-Nb and Al-Ni are 41.0°, 41.5° and 43.2°, respectively. The value of the radius of the first coordination sphere R can be used through the Ehrenfest equation:(1)Q=4πsinθ/λ=1.23(2π/R),
where Q is the scattering vector, 2θ is the scattering angle, λ is the radiation wavelength.

Naturally, the R can be derived from Equation (1) as:(2)R=1.23λ∕2sinθ.

There is an inverse relationship between R and θ. Generally, the R was determined by the sizes of the alloy elements in the system. Hence, with adding Zr (0.162 nm), Nb (0.143 nm) and Ni (0.125 nm), the value of R decreases and the 2θ position of the diffuse maxima of the amorphous alloys in the XRD patterns towards larger angles.

The XRD pattern of the three studied amorphous alloys all exhibit asymmetry of the main part of the diffuse maxima. Especially, the analysis of the XRD pattern of Al-Zr has shown that there are two typical broad diffuse diffraction peaks, the pre-peak diffuse maxima at 32.4° and the main diffuse maxima at 41.0°. This finding can be inferred in the Al-Zr amorphous alloys [20]. Appearance of a shoulder or developing asymmetry of the main diffuse maxima indicates that the amorphous alloys contains the areas with different shortest interatomic distances [23], as apparently form different types of amorphous structure.

To verify the amorphous structure, the TEM and SAED patterns of Al-Zr and Al-Ni alloys powders were shown in Figure 2.

There is no visible crystals in the TEM and SAED patterns of Al-Ni, indicating the complete formation of amorphous. However, there are ordered lattice fringes in TEM images of Al-Zr. In the SAED pattern, the diffuse halo, rings and diffraction spots all be appeared. This confirms the structure of Al-Zr is not completely amorphous.

The preference for a particular type of amorphous structure is controlled by the effective atomic size ratio between the solute and solvent atoms, R*. With decreasing R*, the preferred polyhedral type changes from the Frank–Kasper type (for R* > 1.2) to the icosahedral type (R ≈ 0.902) and then to the bi-capped square Archimedean antiprism type (R ≈ 0.835) [7]. In comparison, the icosahedral type is the dominant topological in the amorphous alloys. Adding elements with different atomic radii to Al based amorphous alloy will change the R* value. It can be concluded that the R* decreases in the order of the Al-Zr (1.133), Al-Nb (1.000), Al-V (0.923), Al-Ni (0.874), Al-Fe (0.867). The polyhedral type of Al-Fe-V-X (Zr, Nb, Ni) amorphous alloys are different in geometry with the addition of alloy elements brings about the change of the R*. We consider using the average radius of solvent atoms $\bar{r}\;(\bar{r}=\sum_{i=1}^{n}c_{i}r_{i})\;$to obtain the R* of alloys, where $c_{i}$ is the atomic concentration of solvent atoms, $r_{i}$ is the solvent atoms radii. With adding Zr, Nb and Ni, the R* were decreased from Al-Zr (0.934) to Al-Nb (0.912) and then to Al-Ni (0.892). The addition of alloy elements with Nb and Ni were more favorable for the R* to be close to 0.902 in these Al-base amorphous alloys, which is conducive to the formation of icosahedral structure, thus, promoting the formation of amorphous.

In order to study the adding of elements on the properties of amorphous, the DSC of the sample alloys at the heating rates of 20 K/min were carry out. The signal of heat flow change at T_g_ of Al-Ni is much clearer than that of other samples. It should be noted that the exothermic crystallization events could be observed in Figure 3, which confirmed that the transformations from amorphous to crystalline in the alloys. May be due to the structure of Al-Zr is not completely amorphous, its thermal performance of DSC is not obvious. Hence, the thermal properties of Al-Nb and Al-Ni alloys were discussed.

The thermal properties of the glass transition temperature (T_g_), the onset crystallization temperature (T_x_), the liquids temperature (T_l_), the heat capacity step (∆C_p_, the heat capacity difference between the supercooled liquid and glass at T_g_) by NETZSCH Proteus software program are listed in Table 1.

Obviously, the T_g_, T_x_, ∆C_p_ of Al-Ni amorphous alloys are higher than Al-Nb amorphous alloys. The higher T_g_ and T_x_ means that the amorphous state can remain stable to a higher temperature and has higher the resistance to crystallization. The glass transition of metallic glass is the falling out-of-equilibrium of all of transitional motions in liquid metals accompanied by a ∆C_p_. In the metallic glass formers, the ∆C_p_ value equal to a constant 3/2 R (12.5 JK^−1^ mol^−1^, R is gas constant). The ∆C_p_ value of the Al-Nb is almost 3/2 R, which is consistent with this view. However, in Al-Ni amorphous alloys, the value of ∆C_p_ is close to 2 R. In paper [24], it is found that the value of ∆C_p_ is distributed around 3/2 R with error of 10%. Meanwhile, there putted understanding that the ∆C_p_ should vary with the composition in glass forming liquid system [25]. Therefore, the ∆C_p_ should be related to the composition of amorphous alloys via mechanical alloying.

In order to compare the efficiency of the stability of the amorphous state and the GFA of amorphous alloys, the currently proposed GFA criteria [26,27]—the supercooled liquid region ΔT_x_ (ΔT_x_ = T_x_ − T_g_), the reduced glass transition temperature T_rg_ (T_rg_ = T_g_/T_l_), the γ value (γ = T_x_/(T_g_ + T_l_)) and the γ_m_ value (γ_m_ = (2T_x_ − T_g_)/T_l_)—are evaluated. The ΔT_x_ exhibits values of 117.2 K and 88 K for the compositions of Al-Nb and Al-Ni, respectively. It is noteworthy that the larger value of ΔT_x_ indicates that the undercooled liquid can remain stable in a wide temperature range, which is beneficial to hot working but it does not necessarily exhibit the higher ability of GFA. The parameter T_rg_, γ and γ_m_, which has been confirmed to have a better correlation with GFA than ΔT_x_. Through the comparison of T_rg_, γ and γ_m_ one can say that the Al-Ni amorphous alloys exhibit higher stability of amorphous state and GFA in comparison with Al-Nb amorphous alloys.

To make further understand the stability of amorphous state and GFA, the mixing enthalpy $\Delta H_{mix}$($\Delta H_{mix}=\sum_{i=1,i\;\neq \;j}^{n}\Omega_{ij}c_{i}c_{j}$) could be used. Here, $\Omega_{ij}=4\Delta H_{ij}$, $\Delta H_{ij}$ is the mixing enthalpy of the *i*th and *j*th elements [8]. Al is determined to be the main element and the values of Δ*H_Al-j_* (*j* = Fe, V, Zr, Nb, Ni) are −11, −16, −44, −18, −22 kJ/mol, respectively [2]. The values of $\Delta H_{mix}$ were calculated for the Al-Zr (−16.85 kJ/mol), Al-Nb (−12.83 kJ/mol) and Al-Ni (−13.47 kJ/mol) to be negative.

Finally, we focus on the effective atomic size ratio and the mixing enthalpy to enhance the stability of amorphous state and GFA. Zr-base alloys has high GFA and atomic size difference with Al. The largest effective atomic size ratio and negative enthalpy of mixing is Al-Zr alloy in the system. The purpose of the Zr addition is to enhance the stability of amorphous state and GFA. However, the state of Al-Zr is not completely amorphous by analysis of SAED and DSC. In contrast, the effective atomic size ratio of the Al-Nb and Al-Ni alloys will lead to the formation of the icosahedral type with enhance GFA. It is reasonable to consider the main reason of amorphous formation is due to the effective atomic size ratio. In addition, Al-Ni alloys exhibit larger T_g_ and T_x_ compared to Al-Nb alloys. Normally, a larger negative value of $\Delta H_{mix}$ in the amorphous alloys is helpful to enhance the stability of amorphous state and GFA [28]. Beyond that, it should be noted that the difference of atomic radius of alloy elements. The atomic radii size of solvent atoms (Fe, V and Ni) is smaller than the solute atoms (Al) in Al-Ni alloys. Thus, the structure of the solvent atom radii less than the solute atom radius should be beneficial to improving the compactness of the cluster structure, hindering the movement of the atom and improving the stability of the amorphous and GFA.

In order to study the thermal stability of Al-Ni alloys, the DSC curves of the Al-Ni amorphous alloy at the heating rates of 10, 20, 30 and 40 K/min are shown in Figure 4. The results indicate that the DSC curves exhibit glass transition, followed by a supper-cooled liquid region, then two exothermic crystallization events and finally melting. The T_g_, T_x_, T_l_ are marked by arrows in Figure 4 and are listed in Table 2 together with the calculated values of ΔT_x_, T_rg_, γ, γ_m_ in Figure 4. It is clear that the T_g_, T_x_, T_p1_, T_p2_, T_l_ of Al-Ni move to high temperature region with the increase of heating rate. Hence, the thermal properties of Al-Ni amorphous alloys is closely related to the increase of heating rate and influenced by kinetic factors.

The activation energies of crystallization E_x_ corresponding to T_x_ of the Al-Ni amorphous alloy can be estimated and listed in Table 3 by the equations of Kissinger (Equation (3)) and Ozawa (Equation (4)):(3)$$\mathrm{Ln}\;\left(\beta /T^{2}\right)=-\left(E/\mathrm{RT}\right)+\mathrm{Const}$$
(4)Ln (β)=−(E/RT)+Const.Here, the β is the heating rates, T represents the specific temperature T_x_, E is the corresponding activation energy E_x_ and R is the gas constant.

The curves fitted by the least square method are close to the straight lines, the Adjusted R-square of the fitted curves are higher than 98 and the results of the Kissinger and Ozawa have the same tendency which proves the reliability of the data in Figure 5. The values and standard errors of Al-Ni by Kissinger equation or Ozawa equation are 47.8, 3.8 and 49.6, 3.8, respectively. By formula calculation, it is clear that the E_x_ value of Al-Ni is higher than that of Al-Nb by Kissinger equation or Ozawa equation. It indicates that higher energy barrier is needed for the Al-Ni crystallization, which is agreement with the higher stability of amorphous state and GFA in comparison with Al-Nb amorphous alloys.

## 4. Conclusions

Using mechanical alloying, we prepared Al-Zr and Al-Ni alloys to compare with Al-Nb. Micro-structure of alloys were studied by XRD, TEM, SAED and DSC indicated that Al-Zr alloy is not completely amorphous and Al-Ni were amorphous alloys. The Al-Ni amorphous alloys exhibits values of ΔT_x_ was 88 K at the heating rates of 20 K/min and exhibits higher GFA and E_x_ in these alloys. The main reason of GFA be enhanced with Ni addition is attributed to the effective atomic size ratio is conducive to the formation of icosahedral structure and the solvent atom radii less than the solute atom radius in alloy system. These results have potential applications for Al-base amorphous alloys in the fields of powder metallurgy industry, bulk amorphous and nanomaterials.

## Figures and Tables

**Figure 1 nanomaterials-11-00488-f001:**
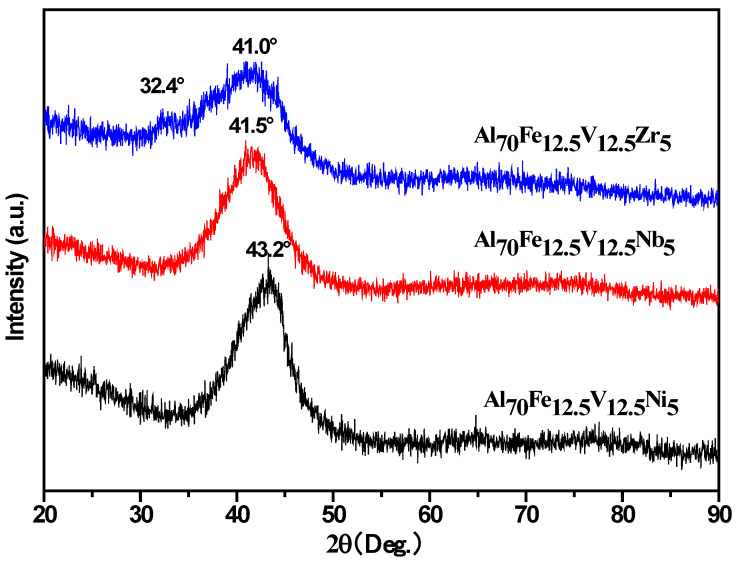
The X-ray diffraction (XRD) patterns of Al-Zr, Al-Nb and Al-Ni alloys milled for 100 h, 60 h and 50 h, respectively.

**Figure 2 nanomaterials-11-00488-f002:**
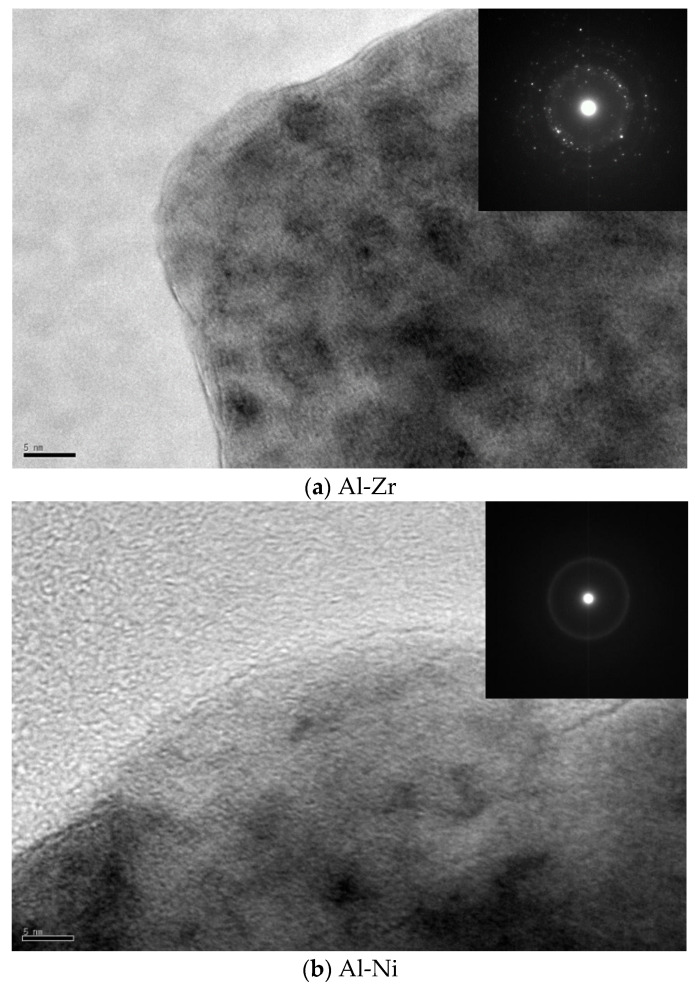
The transmission electron microscopy (TEM) image and selected area electron diffraction (SAED) patterns of Al-Zr and Al-Ni alloy powders.

**Figure 3 nanomaterials-11-00488-f003:**
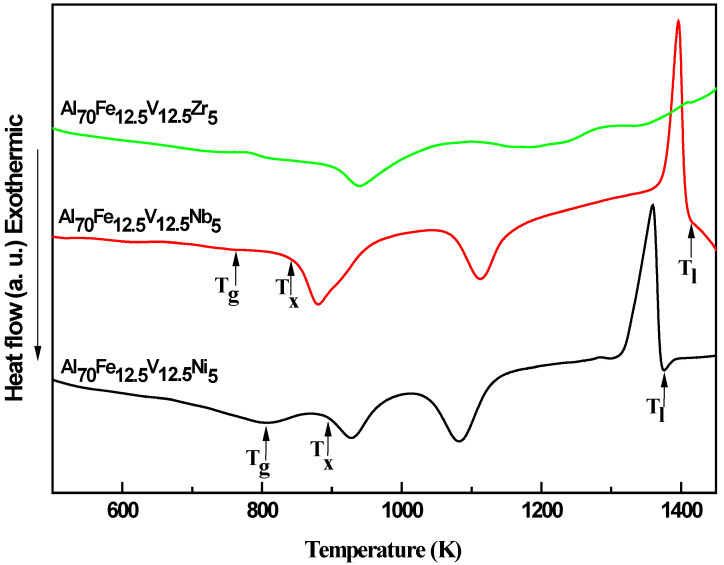
Differential scanning calorimetry (DSC) curves of the Al-Zr, Al-Nb and Al-Ni alloys at the heating rates of 20 K/min.

**Figure 4 nanomaterials-11-00488-f004:**
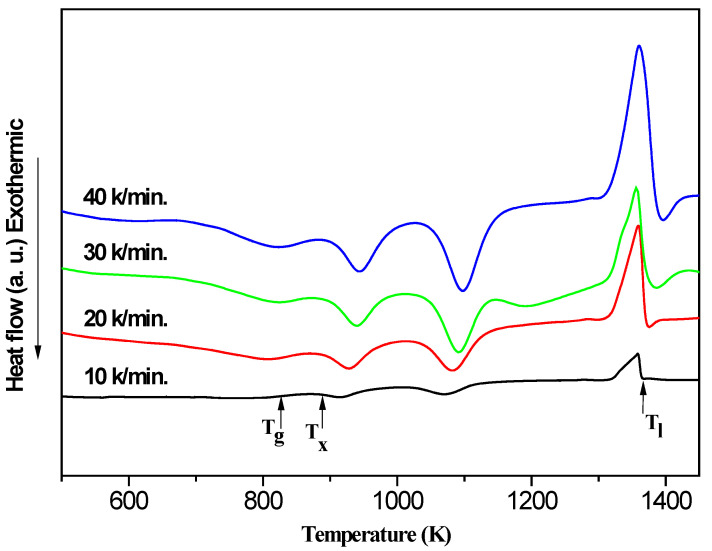
DSC curves of the Al-Ni amorphous alloy at the heating rates of 10, 20, 30 and 40 K/min.

**Figure 5 nanomaterials-11-00488-f005:**
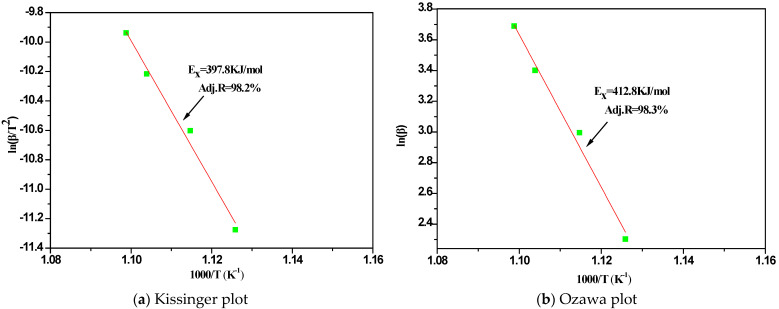
The Kissinger and Ozawa plots of the Al-Ni amorphous alloy.

**Table 1 nanomaterials-11-00488-t001:** The thermal properties and ΔT_x_, T_rg_, γ of alloys measured at 20 K/min heating rates.

Alloys	T_g_ (K)	T_x_ (K)	∆C_p_ (JK^−^^1^ mol^−^^1^)	T_l_ (K)	ΔT_x_ (K)	T_rg_	γ	γ_m_
Al-Nb	736.9	854.1	12.1	1407.9	117.2	0.522	0.398	0.690
Al-Ni	809.1	897.1	16.2	1370.4	88.0	0.590	0.412	0.719

**Table 2 nanomaterials-11-00488-t002:** The thermal properties and ΔT_x_, T_rg_, γ, γ_m_ of Al-Ni measured at 10, 20, 30 and 40 K/min heating rates.

β (K/min)	T_g_ (K)	T_x_ (K)	T_l_ (K)	ΔT_x_ (K)	T_rg_	γ	γ_m_
**10**	797.3	888.2	1369.2	90.9	0.582	0.410	0.715
**20**	809.1	897.1	1370.4	88.0	0.590	0.412	0.719
**30**	819.5	905.9	1373.3	86.4	0.597	0.413	0.723
**40**	827.3	910.1	1381.5	82.8	0.599	0.412	0.719

**Table 3 nanomaterials-11-00488-t003:** Activation energies of crystallization E_x_ estimated with the Kissinger and Ozawa methods.

Equations	Kissinger	Ozawa	Ref.
Al-Nb	366.3 ± 23.9	380.5 ± 23.9	[22]
Al-Ni	396.7 ± 31.5	411.7 ± 31.5	

## Data Availability

The data presented in this study are available on request from the first author.

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
