# Peer review of "Glass-Forming Ability and Thermal Properties of Al70Fe12.5V12.5X5(X = Zr, Nb, Ni) Amorphous Alloys via Minor Alloying Additions"

_nanomaterials, 2021, doi:10.3390/nano11020488_

Round 1

Reviewer 1 Report

The authors study the glass forming ability of 3 Al-Fe-V-based alloys doped by mechanical alloying, with Zr, Nb and Ni. They find that Zr-doping is not able to induce the complete amorphization whereas Ni shows greater glass forming ability than Nb. The authors use x-ray diffraction, differential scanning calorimetry and transmission electron microscopy. While overall the manuscript appears to be interesting, I have some fundamental criticisms and other comments that, in my opinion, must be properly addressed before considering the manuscript worthy for publication.

My main concern is about the DSC results. On the one hand, as a usual procedure, the temperatures for the peaks are usually determined as the onset of the peak, but in Fig. 3 the authors determine the peak temperature T_l as the finishing temperature (as the ramps have been perfomed on heating). Moreover, here it seems that there is some exothermic process just after the peak, could the authors comment on this? Also, in Fig. 4 the peak temperatures Tp1 and Tp2 correspond to the maximum of the peak, and it is widely known that peak maximums in heat flow shift to higher temperatures when the temperature rate increases due to the fact that most DSC devices read temperature from the reference.  So I have serious doubts about the physical meaning of Figs 5(a,b) for the fits performed on Tp1 and Tp2. Also, it should be fine to give details on how Tx have been determined, as it would contain all the relevant physics associated with the corresponding peak. Also, I do not know how the authors have managed to determine Tg for the Nb alloy from the DSC curve (Fig. 3 again), as the peak is undetectable (there are other bumps at lower temperatures in the same curve that are more pronounced that any possible change occurring at Tg). Also, it would be fine to know how the increment in Cp is determined (please provide baselines). Finally, at sight the values given in tables 1 and 2 seem to be given with much larger accuracy than they should, given the DSC curves appearing in the corresponding figures. Please give the corresponding errors (1 significant figure) and give all values with the appropriate number of significant figures. In table 3, errors are given but three significant figures for them seem to be too much. In summary, the procedure to extract information from the DSC measurements should be much more explained.

Other comments:

- In the experimental section the authors say that they have conducted transmission electron microscopy experiments, but I could not find any TEM image nor any reference to these outcomes throughout the manuscript.

- The way figure 1 is presented does not appear to be appropriate. A rigorous comparison should show the results for identical milling times in order to show evidence that lower milling times are needed for the first alloy. On the other hand, how the milling times have been selected a priori?

- GFA should be defined the first time “Glass forming ability” is used in the introduction. On the other hand, I would not use the abbreviation at all in the abstract.

- SEAD is used instead of SAED in some places, please correct.

- I recommend the use of abbreviated names for each alloy, as this would make the reading easier.

- The introduction should emphasize the leitmotiv of this study. Is it important for applications? (now a mention to this issue appears only in the conclusions and without any reference) Is it important for fundamental physics? Why?

Author Response

Response to Reviewer 1 Comments

Dear reviewer:

We are very grateful to your comments for the manuscript. Those comments are all valuable and very helpful for revising and improving our paper, as well as the important guiding significance to our researches. According with your advice, we amended the relevant part in manuscript. Some of your questions were answered below.

Point 1: My main concern is about the DSC results. On the one hand, as a usual procedure, the temperatures for the peaks are usually determined as the onset of the peak, but in Fig. 3 the authors determine the peak temperature T_l as the finishing temperature (as the ramps have been perfomed on heating). Moreover, here it seems that there is some exothermic process just after the peak, could the authors comment on this? Also, in Fig. 4 the peak temperatures Tp1 and Tp2 correspond to the maximum of the peak, and it is widely known that peak maximums in heat flow shift to higher temperatures when the temperature rate increases due to the fact that most DSC devices read temperature from the reference.  So I have serious doubts about the physical meaning of Figs 5(a,b) for the fits performed on Tp1 and Tp2. Also, it should be fine to give details on how Tx have been determined, as it would contain all the relevant physics associated with the corresponding peak. Also, I do not know how the authors have managed to determine Tg for the Nb alloy from the DSC curve (Fig. 3 again), as the peak is undetectable (there are other bumps at lower temperatures in the same curve that are more pronounced that any possible change occurring at Tg). Also, it would be fine to know how the increment in Cp is determined (please provide baselines). Finally, at sight the values given in tables 1 and 2 seem to be given with much larger accuracy than they should, given the DSC curves appearing in the corresponding figures. Please give the corresponding errors (1 significant figure) and give all values with the appropriate number of significant figures. In table 3, errors are given but three significant figures for them seem to be too much. In summary, the procedure to extract information from the DSC measurements should be much more explained.

Response 1:Thank you for the reviewer’s kind advice. The thermal properties of  amorphous alloys is closely related to the increase of heating rate by DSC.As a result, many criteria have been proposed to reflect relative the glass-forming ability (GFA) on the basis of the characteristic temperatures measured by DSC to reduce the influence of test conditions on the amorphous GFA. The values were calculated by the NETZSCH Proteus software program of DSC. Tl is the temperature at which the alloy melts completely. In “Crystallization Behavior of Al70Fe12.5V12.5Nb5 Amorphous Alloy Formed by Mechanical Alloying”(https://doi.org/10.3390/ma12030383), the glass transition region has been enlarged, which can be seen clearly.

we decided to delete the discussion on Tp1 and Tp2 in our manuscript.

The corresponding errors have been revised.

Point 2: In the experimental section the authors say that they have conducted transmission electron microscopy experiments, but I could not find any TEM image nor any reference to these outcomes throughout the manuscript.

Response 2:To address this, we added TEM image shown in Fig. 2 .

Point 3:- The way figure 1 is presented does not appear to be appropriate. A rigorous comparison should show the results for identical milling times in order to show evidence that lower milling times are needed for the first alloy. On the other hand, how the milling times have been selected a priori?

Response 3: When the amorphous powder is prepared by mechanical alloying, the ball milling time of different alloy systems is different. Too long ball milling time will lead to powder pollution and nanocrystallization, so it needs fixed time sampling analysis to determine the ball milling time.Therefore, I have added the ball milling time reason in the experimental details (The milling process was paused every 10 h to take a small amount of powder out of the cans for analysis to determine the milling time.).

Point4:GFA should be defined the first time “Glass forming ability” is used in the introduction. On the other hand, I would not use the abbreviation at all in the abstract.

SEAD is used instead of SAED in some places, please correct.

I recommend the use of abbreviated names for each alloy, as this would make the reading easier.

Response 4: Thank you for your valuable advice.These have been revised.

Point5:- The introduction should emphasize the leitmotiv of this study. Is it important for applications? (now a mention to this issue appears only in the conclusions and without any reference) Is it important for fundamental physics? Why?

Response5: Al-based amorphous alloys have great prospects in the application of light component materials in aerospace and other fields. Efficient consolidation of mechanically alloyed amorphous powders using advanced techniques,such as spark plasma sintering, hot pressing and extrusion could eliminate the problem of dimensional limitation of Al based bulk glassy alloys.We has done the consolidation of Al-Fe-V amorphous in the early stage. At present, we mainly study the optimum design of the Al-based alloy system. The next step is to select the consolidation method and study its mechanical properties.

 The description of the applications of Al-based amorphous alloy was given in the introduction.

We acknowledge the reviewer’s comments and suggestions very much, which are valuable in improving the quality of our manuscript.

Reviewer 2 Report

In the present work of the influence of Zr, Nb, or Ni addition on the glass-forming ability of Al-Fe-V amorphous alloys were studied.  The quality and the contents of the submitted paper are so poor that it should be rejected. There are no new practical or theoretical results. There are methodological mistakes in the selection of research techniques and knowledge-based expertise.

It should not be published in the Nanomaterials unless a significant correction is made and the paper is re-write to meet the Journal's standards. The article requires some significant improvements that the authors may wish to consider when submitting the revised manuscript.

  1. It is necessary to complete the information about the milling process conditions, especially the cup capacity.
  2. It is necessary to complete the information on why different milling times were chosen for different alloying additives? Cite the corresponding literature. In the reviewer's opinion, the diverse selection of milling times for different alloying elements is an evident lack of methodical planning of the experiment. The above clearly disqualifies the work in this form.
  3. An important question is. Wouldn't it be better for the good quality of research to have the same milling times for each composition, then the research conducted in this way would show the appropriate direction of the influence of the elements on the alloy after milling.
  4. The influence of the atomic radius of alloying additive on the value of reflection shift should be discussed, but it is correct only for the same milling times.
  5. There is a lack of description and explanation based on literature references about the effect of the milling time on the changes in the positions of the "amorphous hallo".
  6. The milling times should be given in the details section of the experiment.
  7. It is necessary to correct the references, the actual format in the manuscript does not correspond to the journal standard.

Author Response

Dear reviewer:

 We have studied the valuable comments from you carefully, and tried our best to revise the manuscript.

Point 1:It is necessary to complete the information about the milling process conditions, especially the cup capacity.

Response 1:Thank you for the reviewer’s advice. To address this, we added the information about the milling process conditions in experimental details.

Point 2: It is necessary to complete the information on why different milling times were chosen for different alloying additives? Cite the corresponding literature. In the reviewer's opinion, the diverse selection of milling times for different alloying elements is an evident lack of methodical planning of the experiment. The above clearly disqualifies the work in this form.

Response 2: When the amorphous powder is prepared by mechanical alloying, the ball milling time of different alloy systems is different. Too long ball milling time will lead to powder pollution and nanocrystallization, so it needs fixed time sampling analysis to determine the ball milling time.Therefore, I have added the ball milling time reason in  experimental details (The milling process was paused every 10 h to take a small amount of powder out of the cans for analysis to determine the milling time.).

Point 3:- An important question is. Wouldn't it be better for the good quality of research to have the same milling times for each composition, then the research conducted in this way would show the appropriate direction of the influence of the elements on the alloy after milling.

Response 3: Not all alloy systems can produce amorphous by mechanical alloying.

Al-based amorphous alloys have great prospects in the application of light component materials in aerospace and other fields. Hence,we expect to obtain an alloy system with high glass forming ability and thermal stability through the optimization design of alloy system for later application.

Point4:The influence of the atomic radius of alloying additive on the value of reflection shift should be discussed, but it is correct only for the same milling times.

Response 4: Thank you for your  advice. Because amorphous is a kind of short-range ordered and long-range disordered structure, it is generally characterized by cluster structure, so in the study of amorphous structure, we sometimes analyze the diffraction angle of diffuse maxima peak to understand the approximate range of cluster structure size.

 Point5:-There is a lack of description and explanation based on literature references about the effect of the milling time on the changes in the positions of the "amorphous hallo".

Response5: There is no relationship between milling time and the position of "amorphous hallo". In order to avoid understanding the article, “milled for varied times”in the sentence of “Figure 1 shows the XRD patterns of Al-Zr, Al-Nb and Al-Ni alloys milled for varied times” and “ Among them, the milling time of Al-Ni is the least, which only takes 50 hours.” has been deleted.

Point6:- The milling times should be given in the details section of the experiment.

Response6: Thank you for the reviewer’s kind advice. It have been revised.

Point7:- It is necessary to correct the references, the actual format in the manuscript does not correspond to the journal standard.

Response7: I have changed the style of references according to the journal.

We acknowledge the reviewer’s comments and suggestions very much, which are valuable in improving the quality of our manuscript.

Reviewer 3 Report

Dear Authors,

thank you for your manuscript on the GFA in the AlFeV-X-System. You present a very interesting study. I have some questions:

1. Why did you choose „nanomaterials“ as a Journal? You present a very good manuscript, however, I did not find a match with the scope of „nanomaterials“. I would have expected this contribution in „materials“ oder „metals“.

2. Experimental Details
Please give information on the purity of powders. This is a really critical issue for Zr powder. Also check its amount of contamination due to Hf.
Did you check the acutal composition of the samples?

3. Fig. 1, XRX patterns
Your signals look pretty noisy. What measurement time in the XRD did you use?
This makes it a bit tough to judge if there are additional peaks. For example, in the Nickle alloy there may be a spike right left of the A at approx. 64 degrees, and there are other artifacts like that in the blue spectrum (at around 38 degrees)

4. Fig 3: What is the reproducibility of the experiment? I did not see error bars. How many samples of each composition did you measure?

Author Response

Dear reviewer:

We are very grateful to your comments for the manuscript. Those comments are all valuable and very helpful for revising and improving our paper, as well as the important guiding significance to our researches. According with your advice, we amended the relevant part in manuscript. Some of your questions were answered below.

Point 1: Why did you choose “nanomaterials”as a Journal? You present a very good manuscript, however, I did not find a match with the scope of “nanomaterials”. I would have expected this contribution in “materials”oder “metals”.

Response 1:Thank you for the reviewer’s kind advice. We consider that the size of amorphous powder prepared by mechanical alloying is nanometer, and the pressing and sintering of amorphous alloy powder is an important method to prepare bulk amorphous and nanocrystalline materials. Meanwhile, Nanomaterials  is an international and interdisciplinary scholarly open access journal. We are confident that our study in this article will be interesting to the readers of Nanomaterials . Thank you  for your consideration of this manuscript.

Point 2: Experimental Details
Please give information on the purity of powders. This is a really critical issue for Zr powder. Also check its amount of contamination due to Hf.
Did you check the acutal composition of the samples?

Response 2:Your explanation is very correct.Our previous experiments failed because of the low purity of the powder, and then we used Aladdin's high purity metal powder.The purity of the powders has been listed in experimental details.

Point 3:Fig. 1, XRX patterns
Your signals look pretty noisy. What measurement time in the XRD did you use?
This makes it a bit tough to judge if there are additional peaks. For example, in the Nickle alloy there may be a spike right left of the A at approx. 64 degrees, and there are other artifacts like that in the blue spectrum (at around 38 degrees)

Response 3: The XRD was measured by a special laboratory researcher. The scan step size was 0.0334923,and time per step was 29.84. Generally speaking, the XRD  can not completely judge the amorphous structure. The amorphous or amorphous-nanocrystalline structure will be formed when the amorphous powders are prepared by mechanical alloying, which is shown as broad diffuse diffraction peaks in XRD, so SEAD is needed to prove the amorphous structure.

Point4:Fig 3: What is the reproducibility of the experiment? I did not see error bars. How many samples of each composition did you measure?

Response 4: Thanks for the reviewer’s suggestion.

The thermal properties of  amorphous alloys is closely related to the increase of heating rate and influenced by kinetic factors. As a result, many criteria have been proposed to reflect relative the glass-forming ability (GFA) on the basis of the characteristic temperatures measured by DSC to reduce the influence of test conditions on the amorphous GFA. Hence,the thermal properties of powders were studied by  the DSC (NETZSCH DSC 404F3) at heating rates of 10, 20, 30, 40K/min under a continuous flow of Ar atmosphere.

We acknowledge the reviewer’s comments and suggestions very much, which are valuable in improving the quality of our manuscript.

Round 2

Reviewer 2 Report

Dear Authors,
This is a revised manuscript based on the reviewers' comments and suggestions. The authors provided satisfactory answers to the doubts and questions raised by the reviewers, and revised the manuscript accordingly. I think the manuscript has now met the press condition and recommend accept it for publication in Nanomaterials journal.